# Cinnamaldehyde Increases the Survival of Mice Submitted to Sepsis Induced by Extraintestinal Pathogenic *Escherichia coli*

**DOI:** 10.3390/antibiotics11030364

**Published:** 2022-03-09

**Authors:** Isabella F. S. Figueiredo, Lorena G. Araújo, Raissa G. Assunção, Itaynara L. Dutra, Johnny R. Nascimento, Fabrícia S. Rego, Carolina S. Rolim, Leylane S. R. Alves, Mariana A. Frazão, Samilly F. Cadete, Luís Cláudio N. da Silva, Joicy C. de Sá, Eduardo M. de Sousa, Waldir P. Elias, Flávia R. F. Nascimento, Afonso G. Abreu

**Affiliations:** 1Laboratório de Patogenicidade Microbiana, Programa de Pós-Graduação em Biologia Microbiana, Universidade Ceuma, São Luís 65075-120, Brazil; bellaafigueiredo@hotmail.com (I.F.S.F.); araujo.lorenabio@gmail.com (L.G.A.); raissa_guara@hotmail.com (R.G.A.); itaynaradutra@hotmail.com (I.L.D.); fabricia_sr@hotmail.com.br (F.S.R.); carolina_silvaa04@outlook.com (C.S.R.); leylanesusy@hotmail.com (L.S.R.A.); frazaoamariana@gmail.com (M.A.F.); samillyfrancad@gmail.com (S.F.C.); luiscn.silva@ceuma.br (L.C.N.d.S.); joicyvet@hotmail.com (J.C.d.S.); 2Programa de Pós-Graduação em Ciências da Saúde, Universidade Federal do Maranhão, São Luís 65080-805, Brazil; john_nyramos@yahoo.com.br (J.R.N.); edmsousa@hotmail.com (E.M.d.S.); flavia.nascimento@ufma.br (F.R.F.N.); 3Programa de Pós-Graduação em Biologia Microbiana, Universidade Ceuma, São Luís 65075-120, Brazil; 4Laboratório de Imunofisiologia, Departamento de Patologia, Universidade Federal do Maranhão, São Luís 65080-805, Brazil; 5Laboratório de Bacteriologia, Instituto Butantan, São Paulo 05503-900, Brazil; waldir.elias@butantan.gov.br

**Keywords:** sepsis, inflammation, *Escherichia coli*, cinnamaldehyde

## Abstract

Several natural products have been investigated for their bactericidal potential, among these, cinnamaldehyde. In this study, we aimed to evaluate the activity of cinnamaldehyde in the treatment of animals with sepsis induced by extraintestinal pathogenic *E. coli*. Initially, the *E. coli* F5 was incubated with cinnamaldehyde to evaluate the minimum inhibitory and minimum bactericidal concentration. Animal survival was monitored for five days, and a subset of mice were euthanized after 10 h to evaluate histological, hematological, and immunological parameters, as well as the presence of bacteria in the organs. On the one hand, inoculation of bacterium caused the death of 100% of the animals within 24 h after infection. On the other hand, cinnamaldehyde (60 mg/kg) was able to keep 40% of mice alive after infection. The treatment significantly reduced the levels of cytokines in serum and peritoneum and increased the production of cells in both bone marrow and spleen, as well as lymphocytes at the infection site. Cinnamaldehyde was able to reduce tissue damage by decreasing the deleterious effects for the organism and contributed to the control of the sepsis and survival of animals; therefore, it is a promising candidate for the development of new drugs.

## 1. Introduction

Sepsis is characterized by a life-threatening organ dysfunction resulting from an unregulated immune response to an infection [1]. It represents a serious global public health problem due to its elevated rate of morbidity and mortality in intensive care units; it is also a difficult and costly treatment [2,3].

*Escherichia coli* is a frequent bacterial agent of sepsis [4]. It is widely distributed and found in the large intestine of humans, as well as warm-blooded animals as part of the microbiota [5]. Virulence factors are encoded by genes restricted to pathogenic *E. coli*, being absent in commensal bacteria, making these microorganism etiologic agents of intestinal and extraintestinal infections, including bacteremia and sepsis [6].

Recently, when studying the function of protein involved in colonization (Pic) in the pathogenesis of sepsis caused by *E. coli*, our group demonstrated that this serine protease was responsible for inducing lethal sepsis in Swiss mice, causing their death in up to 12 h, with a significant increase in cytokine levels and other inflammatory mediators [7]. Pic is produced by *E. coli*, *Shigella flexneri*, and *Citrobacter rodentium* [8,9,10] and has several functions such as hemagglutination, coagulation cascade factor V degradation, mucinolytic activity [8,11], leukocyte surface glycoprotein cleavage [12], intestinal colonization of mice and rabbits [8,13], and cleavage of several molecules belonging to the complement system [14].

According to the World Health Organization, antibacterial drugs have become less effective, resulting in an accelerating worldwide health security emergency [15]. Therefore, due to the difficult treatment of several microbial infections, it is important to identify and characterize natural products that can be used in the treatment of microbial diseases. As an example, cinnamon bark has been widely studied for therapeutic purposes and has an important role for both research and popular use, since it contains a large amount of essential oil. Cinnamaldehyde is the main active constituent of cinnamon essential oil, and the major bioactive compound isolated from the leaves of *Cinnamomum* spp. [16].

Several functions have been described for the compound, such as: anticancer/antitumor activity [17]; cardioprotective effect [18]; anti-inflammatory properties [19,20]; and antimicrobial activity against several pathogens, such as fungi [21] and bacteria [16,22].

Therefore, due to several properties of cinnamaldehyde, especially the antimicrobial and anti-inflammatory activity, the objective of this work was to evaluate the activity of cinnamaldehyde in the treatment of animals with sepsis induced by extraintestinal pathogenic *E. coli*.

## 2. Results

### 2.1. Cinnamaldehyde Increases the Survival Rate of Septic Animals

On the one hand, cinnamaldehyde at a concentration of 30 mg/kg showed no effect in the survival rate in prophylactic or therapeutic treatments. On the other hand, when animals were treated with cinnamaldehyde at a concentration of 60 mg/kg, the survival rate increased by 16.6% in the prophylactic treatment and by 40% in the therapeutic treatments as compared with non-treated infected mice (Figure 1).

After these results, the therapeutic treatment with the dose of 60 mg/Kg was chosen to follow the study and the mice were euthanized ten hours post infection to evaluate the immunological parameters.

### 2.2. Cinnamaldehyde Promotes Bactericidal Effect In Vitro, but Not In Vivo

Using the MIC method, it was shown that cinnamaldehyde inhibited *E. coli* F5 growth (6 mg/mL). In addition, cinnamaldehyde was bactericidal at the concentration of 12 mg/mL.

*E. coli* F5 was present in both bacterial groups (F5 and treated) on the peritoneal lavage, blood, liver, spleen, and lungs, indicating that the mice had systemic bacteremia. In the PBS and cinnamaldehyde controls, as expected, there was no bacterial growth (data not shown). Nevertheless, no significant difference was observed among the infected groups (Figure 2), suggesting that the increase in survival was not related to bactericidal activity of cinnamaldehyde.

### 2.3. Cinnamaldehyde Decreases Tissue Damage in Animals Promoted by E. coli F5

The animals of infected groups presented histopathological alterations typical of inflammation in the lungs, spleen, and liver as compared with the non-infected control groups (PBS and CIN groups), with the presence of edema, hemorrhage, and cellular infiltrate. However, it was possible to observe a moderate reduction in the hemorrhagic process, as well as a reduction in the inflammatory infiltrate in animals infected and treated with cinnamaldehyde in relation to the non-treated infected group (Table 1 and Figure 3).

The presence of congestion in the groups analyzed was due to the euthanasia procedure. Necrosis was not observed in any analyzed organs; however, the presence of mild periportal edema was detected in the liver of most animals in the non-treated infected group.

In summary, the cinnamaldehyde treatment was able to reduce tissue damage, especially in the lungs, contributing to animal survival.

In Figure 3, arrows show inflammatory infiltrated area and asterisks show hemorrhagic area.

### 2.4. Cinnamaldehyde Treatment Promotes an Increase in Leukocytes in the Bone Marrow, Peritoneal Cavity, and Spleen

The infection induced a reduction in the total blood cell count as compared with PBS and CIN non-infected groups, which was related to a decreasing in lymphocytes number. The neutrophils and monocytes were not reduced after infection. The treatment with cinnamaldehyde did not alter the cell profile in the blood in the infected mice (Figure 4).

In the same way, the number of peritoneal cells was decreased in the infected animals as compared with the PBS and CIN non-infected groups. The treatment with CIN induced an increase in the number of peritoneal cells as compared with the non-treated infected group (Figure 5).

The decrease in blood cell and peritoneal cell numbers in the infected mice was not related to a decrease in cell production by bone marrow, considering that no differences among infected and PBS groups were observed in bone marrow cell counts (Figure 6).

It is possible to observe a small reduction in the number of cells in the non-infected group. However, this reduction is not statistically significant. On the other hand, the administration of cinnamaldehyde to infected animals leads to a significant increase in the number of total cells in both bone marrow and spleen (Figure 7) as compared with the non-treated infected group.

### 2.5. Cinnamaldehyde Promotes a Decrease in Cytokine Levels in Serum and Peritoneum

Proinflammatory cytokines (IL-6, IL-12, IFN-γ, and TNF-α), anti-inflammatory cytokine (IL-10), and chemokine (MCP-1) were detected in serum (Figure 8) and peritoneal lavage (Figure 9) from the *E. coli* F5 infected group. The cinnamaldehyde treatment significantly reduced the levels of all these mediators, with the exception of TNF-α in the serum.

## 3. Discussion

We have recently shown that the intraperitoneal inoculation of *E. coli* strain F5 caused the death of 100% of the animals within 24 h after infection [7]. In this work, we demonstrated that a treatment with cinnamaldehyde (60 mg/kg) improved the survival of animals infected with *E. coli* F5 using the same murine sepsis model. This effect was not due to a bactericidal activity, since there was no reduction in bacteria in the blood and organs of animals after cinnamaldehyde treatment, but likely, due to an anti-inflammatory effect indicated by a reduction in systemic inflammatory mediators and decreased lung damage.

Histological analyses from liver, lungs, and kidneys of the animals infected with *E. coli* F5 presented cellular infiltrate, hemorrhage, and edema. However, the treatment with cinnamaldehyde (60 mg/kg) reduced tissue damage. Probably, the animals survived longer due to the decrease in lung inflammation induced by the compound.

As observed here, several studies have also demonstrated protective effects of cinnamaldehyde in controlling infections. Recently, we demonstrated that cinnamaldehyde treatment reduced the intestinal colonization of mice infected by *E. coli* [22]. Yang et al. [23] showed that trans-cinnamaldehyde attenuated the intestinal histological damage in *Cronobacter sakazakii*-infected newborn mice. Tung et al. [24] showed that bioactive phytochemicals of leaf essential oils of *Cinnamomum osmophloeum* significantly reduced the incidence of liver lesions due to lipopolysaccharide/D-galactosamine-induced acute hepatitis. In addition, cinnamaldehyde was able to protect against rat intestinal ischemia/reperfusion injuries [25], reduced the infarct area in cerebral ischemia mouse model [26], as well as exhibited cardioprotective effects against acute ischemic myocardial injury induced by isoproterenol in rats [18].

In this study, the blood cell counts showed a decrease in leucocytes from animals infected with *E. coli* F5 as compared with the other groups not infected, which may have occurred due to a reduction in lymphocytes. Previously, we described that Pic reduced the number of T and B lymphocytes and compromised the expression of several cell surface molecules in leukocytes, such as CD80 and CD86 [7]. Probably, this reduction in blood cell and peritoneal cell numbers was related to a failure in the recruitment caused by Pic, since in the bone marrow the production was not compromised.

Other studies have also shown that lymphocyte apoptosis was increased in septic patients, leading to a persistent and profound lymphopenia [27,28]. Here, we showed that cinnamaldehyde induced an increase in leukocytes in the bone marrow, and spleen, as well as in the peritoneum, the focus of infection. However, treatment with cinnamaldehyde did not alter the cell profile in the blood in the infected mice. The same was observed by Mendes et al. [29] who investigated the effects of cinnamaldehyde in the inflammatory response triggered by LPS injection in mice. The study showed that cinnamaldehyde treatment did not affect the number of total peripheral blood leukocytes or the number of circulating polymorphonuclear cells in LPS-injected mice. However, an evident reduction in the number of circulating mononuclear cells was observed in the same animals.

It is important to mention that neutrophils are important during the initial immune response to eliminate the microorganisms in the sepsis [30]. As previously demonstrated by Dutra et al. [7], the production of neutrophils is affected by Pic. Thus, the reduction in neutrophils in the *E. coli* F5 group may also have contributed to the high mortality of animals. Moreover, cinnamaldehyde was also not able to induce an increase in the number of neutrophils in serum. In fact, even with an increase in the number of cells in the peritoneum, the treatment with cinnamaldehyde promoted a reduction in the level of cytokines in the peritoneal lavage.

We demonstrated that Pic induced increased production of cytokines, since mice infected with *E. coli* F5 significantly produced more cytokines than its defective Pic mutant [7]. Other studies have also described increased production of cytokine in murine model of sepsis, such as IL-6, as well as the chemokine MCP-1 [29,31,32,33]. Here, when animals with sepsis were treated with cinnamaldehyde, it was possible to observe a reduction in the levels of all cytokines in both serum and peritoneum, with the exception of TNF-α in the serum. Interestingly, there was a reduction in the level of anti-inflammatory cytokine IL-10, suggesting that cinnamaldehyde induces an overall suppression in the immune response.

According to our data, other studies have described the anti-inflammatory effect of cinnamaldehyde in reducing the release of several cytokines. Pannee et al. [34] showed that, in a concentration-dependent manner, cinnamaldehyde significantly inhibited the expression of IL-6, IL-1β, and TNF-α in activated J774A.1 cells. Liao et al. [20] reported that cinnamic aldehyde reduced TNF in LPS-induced murine macrophages, as well as in a carrageenan-induced paw edema model, demonstrating that the compound had excellent anti-inflammatory activities. Mendes et al. [29] demonstrated that cinnamaldehyde also significantly diminished TNFα in plasma.

It is worth noting that the reduction in the level of inflammatory cytokines is important, since exacerbated increased levels of cytokines cause a systemic disorder, which can lead to death. It has already been reported that high concentrations of MCP-1 have also been linked to mortality in sepsis in humans. Zhu et al. [35] reported that plasma levels of MCP-1 were significantly higher in a non-survivor as compared with a survivor group. Mera et al. [36] also showed that the expression levels of MCP-1 soon after intensive care unit admission were higher in non-survivors as compared with survivors.

## 4. Materials and Methods

### 4.1. Bacterial Strain

*E. coli* strain F5, an extraintestinal pathogenic *E. coli* (ExPEC) isolated from the bloodstream [7], was used in this study as the sepsis inducing bacterium. The strain was aerobically grown at 37 °C for 18 h in Luria-Bertani (LB) broth, MacConkey, or LB agar. The strain was kept in LB broth supplemented with 15% glycerol at −80 °C until experimental procedures.

### 4.2. Minimum Inhibitory (MIC) and Minimum Bactericidal Concentration (MBC)

*E. coli* F5 was cultivated in Mueller Hinton broth and grown until it reached an optical density of 0.1, determined at 600 nm. A total of 10 μL of this suspension were added to wells containing cinnamaldehyde at concentrations ranging from 200 to 2500 μg/mL. After 24 h incubation at 37 °C, resazurin 0.03% was used to determine the MIC.

Aliquots from all the wells which showed no visible bacterial growth were seeded on MacConkey agar plates and incubated for 24 h at 37 °C to determine the MBC [37]. All assays were performed in triplicate with at least two repetitions.

Cinnamaldehyde (trans-cinnamaldehyde 99%) used in this study was obtained commercially from Sigma-Aldrich (Darmstadt, Germany). To carry out the experiments, 20% dimethyl sulfoxide (DMSO) was used as solvent.

### 4.3. Animals

Six-to-eight-week-old female Swiss mice, weighing ~25 g, were used in the study. The animals were obtained from the Central Animal House of the Ceuma University (São Luís, Brazil) and maintained at 26 + 2 °C, 44% to 56% relative humidity, under 12 h light-dark cycles and maintained with free access to sterile food and water.

#### 4.3.1. Experimental Design

A total of 42 animals were distributed into seven experimental groups (*n* = 6): two control groups, without infection, which received only phosphate buffered solution (PBS) or cinnamaldehyde (CIN 60 mg/kg); a third group infected with *E. coli* F5 and did not receive any treatment (*E. coli* F5); two prophylactic groups that received cinnamaldehyde at 30 mg/kg (prophylactic CIN 30) or 60 mg/kg (prophylactic CIN 60) for 5 days before infection; finally, two therapeutic groups that received cinnamaldehyde at 30 mg/Kg (therapeutic CIN 30) or 60 mg/kg (therapeutic CIN 60) 2 h after infection. The animals were infected with suspensions containing 10^9^ CFU/mL *E. coli* F5 (200 μL) inoculated intraperitoneally. All the treatments were performed by gavage.

The survival of the animals was evaluated every 12 h until the 5th day after infection [31], and after this period of time, the mice were euthanized with an overdose of the anesthetic (150 mg/kg ketamine hydrochloride and 120 mg/kg xylazine hydrochloride).

After survival analysis, it was possible to observe that the therapeutic CIN 60 group survived for longer periods. Then, another 24 animals were divided into 4 groups: PBS, CIN, *E. coli* F5, and therapeutic CIN 60, which were infected and treated, as described above.

#### 4.3.2. Blood Sampling

After ten hours of infection, the animals were anesthetized to collect the blood with a solution of 2% xylazine chloride (20 mg/kg) and 5% ketamine chloride (25 mg/kg) at a 2:1 ratio [38] via an intramuscular injection. The blood was used to determine differential blood cell counts, nitric oxide (NO) and cytokine levels, as well as to quantify the presence of *E. coli* F5. Blood cell counts were performed using Bioclin 2.8 Vet (Bioclin, Belo Horizonte, Brazil).

#### 4.3.3. Bone Marrow, Spleen, and Peritoneal Cell Counting

After euthanasia, animal peritoneal cells were collected by washing the peritoneal cavity with 5 mL sterile ice-cold PBS. Then, the femur was removed and perfused with 1 mL of PBS to obtain bone marrow cells. Finally, the spleen, liver, kidneys and the lungs were removed. The spleen was crushed with 5 mL PBS and passed through a silk sieve. Total cell number counting was performed according to Cruz et al. [39] using a hemocytometer with the aid of an optical microscope at 400× magnification.

#### 4.3.4. Colony Forming Units (CFU) Counting

To quantify the number of bacteria in the organs, serial dilutions from the peritoneal lavage, blood, and macerated organs (liver, lungs, spleen, and kidneys), were cultivated on MacConkey agar at 37 °C. After overnight incubation, CFUs were counted and the results were expressed as Log of CFU/mL, as described by Dutra [7].

#### 4.3.5. Histopathological Evaluation

The collected samples (kidney, liver, and lung) were fixed in 10% formaldehyde solution for 24 h. They were sectioned and processed for inclusion in paraffin. Sections with 5 mm were stained in hematoxylin-eosin for histopathological analysis. The following parameters were evaluated: edema, necrosis, cellular infiltrate, and hemorrhage. The classification (0 = absent; 1 = slight; 2 = moderate; 3 = intense) was performed, according to that described by Liberio et al. [40].

#### 4.3.6. Cytokines Quantification

Dosage of MCP-1, IFN-γ, TNF-α, IL-12, IL-10, and IL-6 in serum and peritoneal lavage from animals was performed by flow cytometry, using the Mouse Inflammation Cytokine kit (Becton Dickinson Biosciences, San Jose, CA, USA), according to manufacturer’s instructions.

### 4.4. Statistical Analysis

Statistical analyses were performed by using the GraphPad Prism software, version 8.0. The results were expressed as the mean ± standard deviation, and were subjected to ANOVA, followed by T-test and Tukey’s multiple test, or Kruskal–Wallis and Mann–Whitney tests when the data normality assumption was not satisfied. The Kaplan–Meier curve and the log-rank test were used for survival analysis. The differences were considered to be significant when *p* < 0.05.

### 4.5. Ethics Approval

The experimental procedures were carried out under approved guidelines of the National Institutes of Health (NIH) Guide for the Care and Use of Laboratory Animals. All procedures were approved by the Committee of Ethics in Research at the Ceuma University (process no. 22/18).

## 5. Conclusions

Taken together, our data point to an important immunomodulator effect of cinnamaldehyde on the sepsis caused by extraintestinal pathogenic *E. coli*. The compound was also able to reduce the levels of cytokines IL-6, IL-10, IL-12, TNF-α, and IFN-γ, as well as the chemokine MCP-1 in serum and peritoneum. In addition, cinnamaldehyde increased the production of cells in both bone marrow and spleen, and the lymphocyte number at the infection site. Finally, it was able to reduce the inflammatory infiltrate in animals infected and treated in relation to non-treated infected group, decreasing the deleterious effects for the organism and contributing positively to the control of the sepsis and survival of animals.

The compound is efficient in the treatment of the sepsis caused by extraintestinal pathogenic *E. coli* and a promising candidate for the development of new drugs that can be used in the treatment of infections.

## Figures and Tables

**Figure 1 antibiotics-11-00364-f001:**
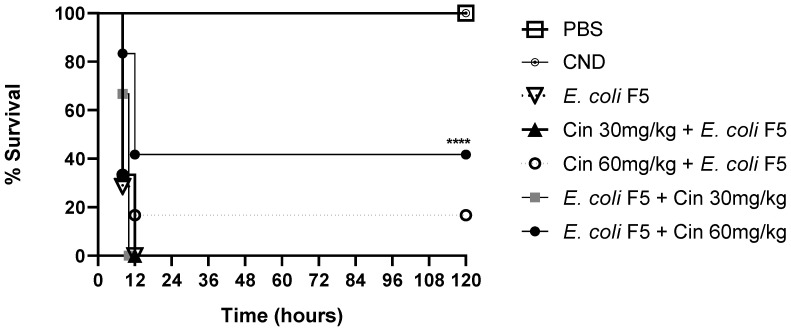
Survival curve of animals infected with *E. coli* F5 and cinnamaldehyde treatment. The mice used in the experiment were distributed into seven groups (*n* = 6). The animals in the non-infected control groups received PBS or cinnamaldehyde (CIN 60 mg/kg). The remaining 3 groups were infected with *E. coli* F5. Prophylactic cinnamaldehyde groups received 30 (prophylactic CIN 30) and 60 mg/kg (prophylactic CIN 60) of cinnamaldehyde, 5 days before the infection, and therapeutic cinnamaldehyde groups were treated with 30 (therapeutic CIN 30) and 60 mg/kg (therapeutic CIN 60) of cinnamaldehyde, 2 h after *E. coli* F5 infection. Then, the survival was evaluated every 12 h until the 5th day after infection. **** *p* < 0.0001 as compared with the *E. coli* F5 group.

**Figure 2 antibiotics-11-00364-f002:**
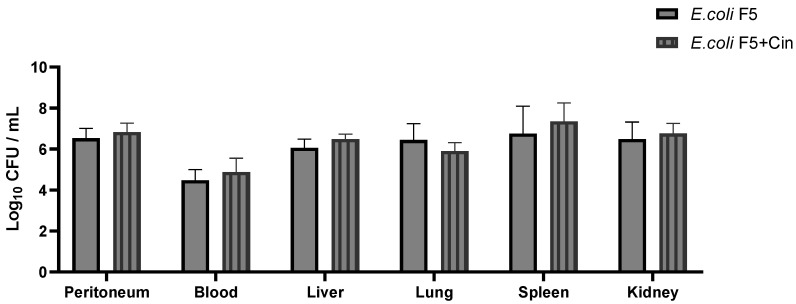
CFU counting from blood, peritoneal lavage, lungs, liver, spleen, and kidneys from animals infected with *E. coli* F5 and cinnamaldehyde treatment. The animals used in the experiment were distributed into 4 groups (*n* = 6). The groups received *E. coli* F5 or *E. coli* F5 followed by cinnamaldehyde treatment 60 mg/kg after two hours. After 10 h, the mice were euthanized. Aliquots of peritoneal fluid and blood were diluted in sterile PBS and serial dilutions were cultivated on MacConkey for evaluation of CFU. To quantify bacteria in the organs, part of them were macerated, and dilutions were cultivated on MacConkey. After overnight incubation at 37 °C, the CFUs were counted, and the results expressed as Log of CFU/mL.

**Figure 3 antibiotics-11-00364-f003:**
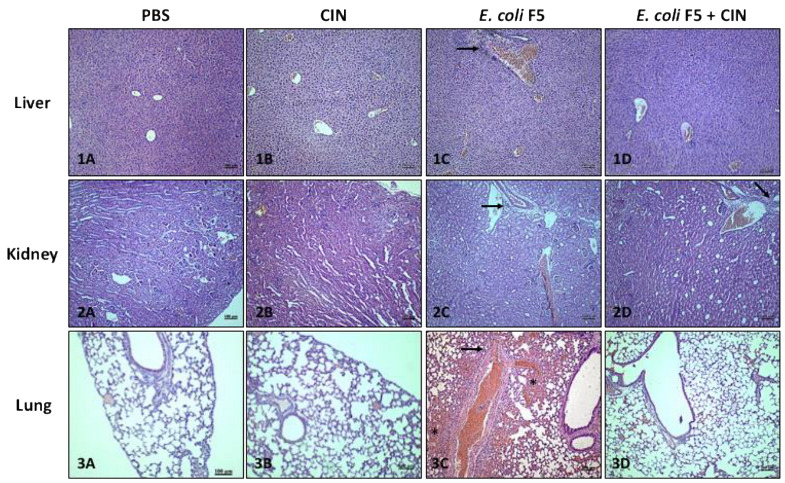
Histopathological analyzes of lungs, liver, and kidneys from animals infected with *E. coli* F5 and cinnamaldehyde treatment. Ten hours after peritoneal inoculation of *E. coli* F5, the organs of the animals were removed for histopathological analysis. Liver: (**1A**) PBS; (**1B**) CIN; (**1C**) *E. coli* F5; (**1D**) *E. coli* F5+CIN. Kidney: (**2A**) PBS; (**2B**) CIN; (**2C**) *E. coli* F5; (**2D**) *E. coli* F5+CIN. Lung: (**3A**) PBS; (**3B**) CIN; (**3C**) *E. coli* F5; (**3D**) *E. coli* F5+CIN. The scale bar is 100 μm.

**Figure 4 antibiotics-11-00364-f004:**
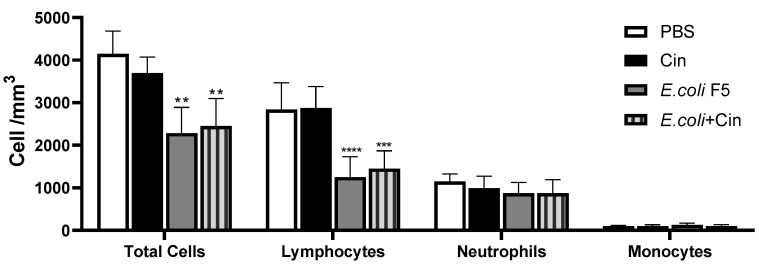
Total and differential blood cell counts from animals infected with *E. coli* F5 and cinnamaldehyde treatment. Ten hours after peritoneal inoculation of *E. coli* F5, blood samples were collected for total and differential cell counting. The results represent the mean ± S.D. of 6 animals/group. ** *p* < 0.01 as compared with the PBS group, *** *p* < 0.001 as compared with the PBS group, and **** *p* < 0.0001 as compared with the PBS group.

**Figure 5 antibiotics-11-00364-f005:**
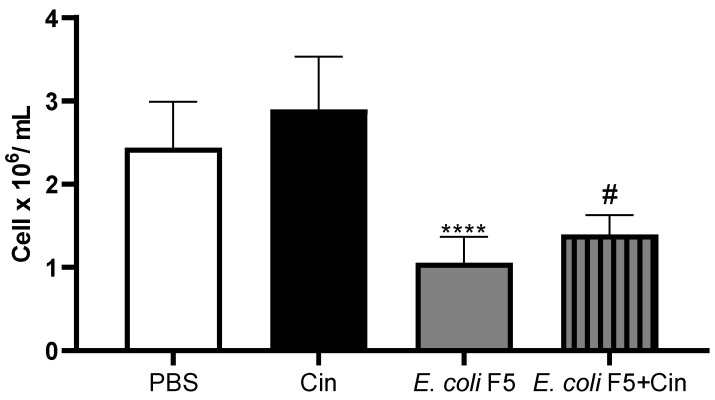
Peritoneal cell counts of animals infected with *E. coli* F5 and cinnamaldehyde treatment. Ten hours after peritoneal inoculation of *E. coli* F5, the peritoneal cells of the mice were collected from the lavage of the peritoneal cavity. The cells were stained with 0.05% violet crystal in 30% acetic acid and counted under a light microscope. The results represent the mean ± S.D. of 6 animals/group. **** *p* < 0.0001 as compared with the CIN group and # *p* < 0.05 as compared with the *E. coli* F5 group.

**Figure 6 antibiotics-11-00364-f006:**
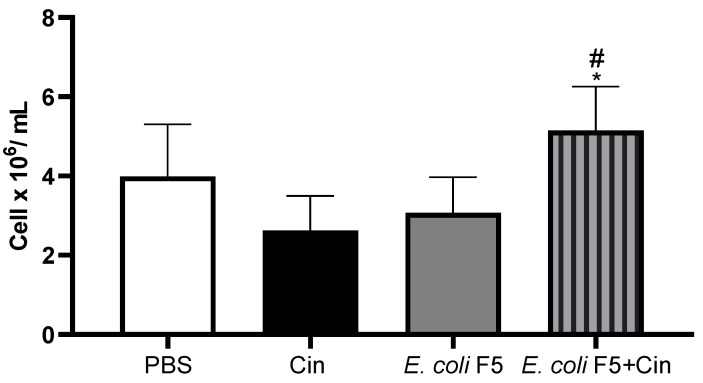
Total bone marrow cell counts from animals infected with *E. coli* F5 and cinnamaldehyde treatment. Ten hours after peritoneal inoculation of *E. coli* F5, bone marrow cells were obtained. The cells were stained with 0.05% violet crystal in 30% acetic acid and counted under a light microscope. The results represent the mean ± S.D. of 6 animals/group. * *p* < 0.05 as compared with the *E. coli* F5 group and # *p* < 0.01 as compared to the CIN group.

**Figure 7 antibiotics-11-00364-f007:**
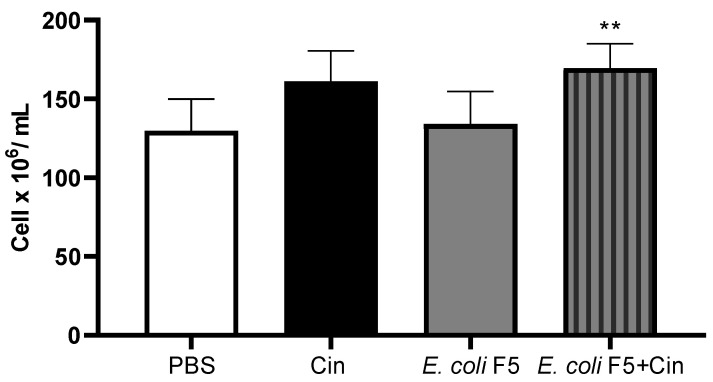
Total spleen cell counts from animals infected with *E. coli* F5 and cinnamaldehyde treatment. Ten hours after peritoneal inoculation of *E. coli* F5, spleen cells were obtained by removing and macerating the organ using PBS. The cells were stained with 0.05% violet crystal in 30% acetic acid and counted under a light microscope. The results represent the mean ± S.D. of 6 animals/group. ** *p* < 0.05 as compared with the *E. coli* F5 group.

**Figure 8 antibiotics-11-00364-f008:**
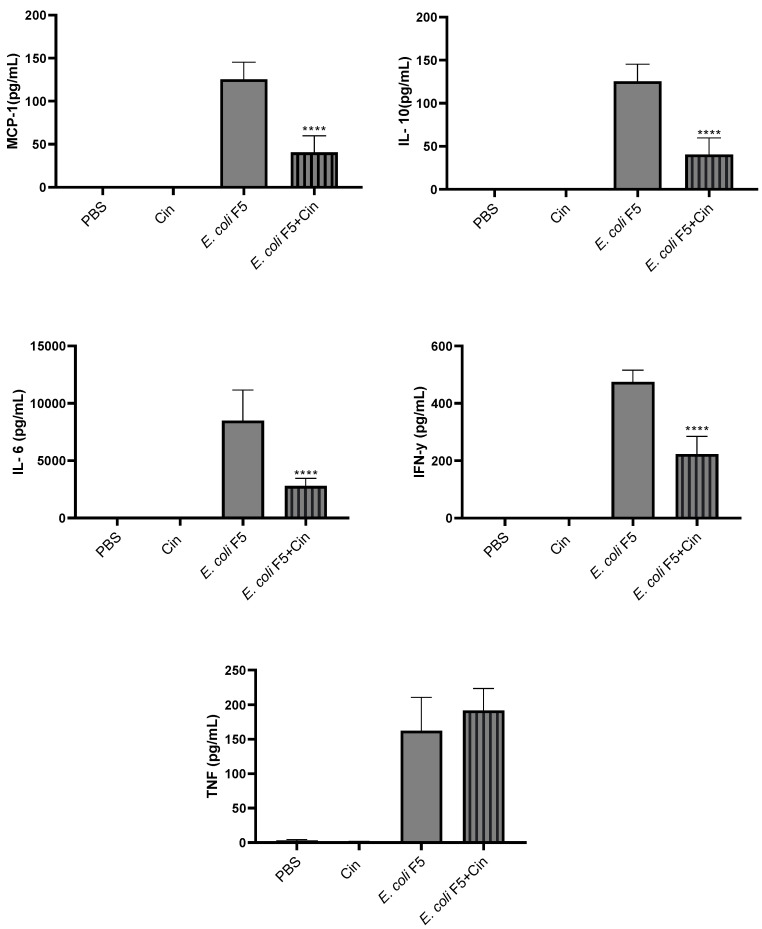
Detection of cytokines in the sera of animals infected with *E. coli* F5 and cinnamaldehyde treatment. Ten hours after peritoneal inoculation of *E. coli* F5, animal sera were used for cytokines dosage by flow cytometry. The results represent the mean ± S.D. of 6 animals/group. **** *p* < 0.0001 as compared with the *E. coli* F5 group.

**Figure 9 antibiotics-11-00364-f009:**
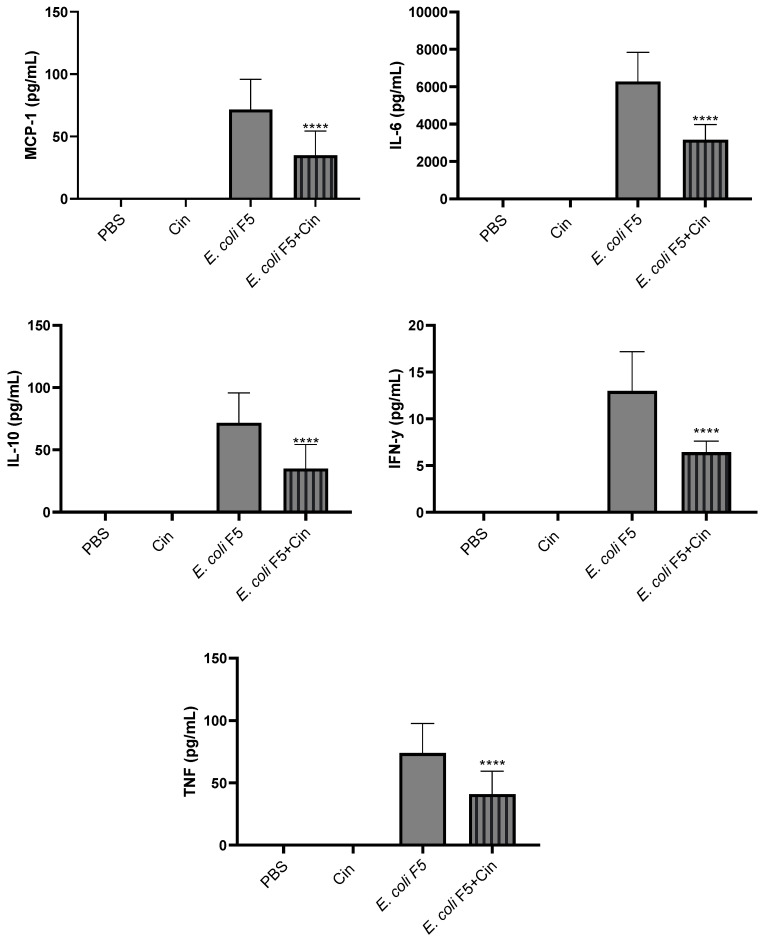
Detection of cytokines in the peritoneal lavage of animals infected with *E. coli* F5 and cinnamaldehyde treatment. Ten hours after peritoneal inoculation of *E. coli* F5, animal peritoneal lavages were used for cytokines dosage by flow cytometry. The results represent the mean ± S.D. of 6 animals/group. **** *p* < 0.0001 as compared with the *E. coli* F5 group.

**Table 1 antibiotics-11-00364-t001:** Histopathological evaluation of kidneys, lungs, and liver from animals submitted to intraperitoneal inoculation of bacteria and cinnamaldehyde treatment.

		Hemorrhage	Infiltrated	Edema	Necrosis
Kidney	PBS	0	0.7 ± 0.5	0	0
	Cin	0	1.0 ± 0.8	0	0
	*E. coli* F5	1.3 ± 0.5 ^a^	1.7 ± 0.5	0	0
	*E. coli* F5+Cin	0.7 ± 0.7	1.7 ± 0.5	0	0
Lung	PBS	0	0.7 ± 0.5	0	0
	Cin	0	1.0 ± 0.0	0	0
	*E. coli* F5	1.3 ± 0.7	2.2 ± 0.4	0	0
	*E. coli* F5+Cin	1.2 ± 0.4	1.8 ± 0.4	0	0
Liver	PBS	0	1.0 ± 0.0	0	0
	Cin	0	1.0 ± 0.0	0	0
	*E. coli* F5	2.8 ± 0.7	2.2 ± 0.4	0.7 ± 0.5	0
	*E. coli* F5+Cin	1.2 ± 0.4	2.0 ± 0.0	0.3 ± 0.5	0

^a^ The results are expressed as mean ± SD of the scores: 0, absent; 1, mild; 2, moderate; 3, intense.

## Data Availability

Data are contained within the article.

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
