# Peer review of "Cinnamaldehyde Increases the Survival of Mice Submitted to Sepsis Induced by Extraintestinal Pathogenic Escherichia coli"

_antibiotics, 2022, doi:10.3390/antibiotics11030364_

Round 1
Reviewer 1 Report
The submitted manuscript entitled “Cinnamaldehyde increases the survival of mice submitted to sepsis induced by extraintestinal pathogenic Escherichia coli reducing lung lesions and systemic inflammation” describes the compound cinnamaldehyde as a promising candidate for the development of new drugs. The methods are clearly described and the paper is of interest to the readers of Antibiotics, and I would recommend accepting the manuscript after the comments below are addressed.
Corrections:
- Line 23: “The aim of this study was” could be revised as “The study aimed to”
- Line 27: The word “ analyzes” should be corrected as “analysis”
- Line 28: The word “was” should be corrected as “were”
- Line 32: The word “cell” should be corrected as “cells”
- Line 34: “…damage, decreasing…” should be corrected as “…damage by decreasing…”
- Line 35: “for the development of”
- Line 41: The word “problem” should be corrected as “problems”
- Line 60: “As an example”
- Line 65: add the word “and” before “anti-inflammatory properties”
- Lines 66-68: What is this sentence trying to describe? Please double-check.
- Line 84: Add a comma between “added” and “and”
- Line 88: “treated with cinnamaldehyde at the concentration of 60 mg/kg” or “treated with the dose”
- Line 101: The word “dosis” should be corrected as “dose”
- Line 125: “with the presence of edema”
- Line 128: “to the non-treated infected group”
- Line 147: “promotes an increase of”
- Line 163: “to the non-treated infected”
- Line 177: The word “cell” should be corrected as “cells”
- Line 178: “to the non-treated infected”
- Line 213: Remove the extra “7”
- Line 222: “cinnamaldehyde for two hours…”
- Line 288: Is there any unit of “ density of 0.1 at 600 nm.”
- Line 289: “in concentrations ranged from” should be corrected as “at concentrations ranging from”
- Line 290: “was added resazurin 0.03% to” should be corrected
- Line 302: “plates were incubated for 3 h”
- Line 318: the sentence “finally, two therapeutic groups that were infected with E. coli F5 and after 2 h 318 received cinnamaldehyde at 30 mg/Kg (Therapeutic CIN 30) or 60 mg/kg (Therapeutic 319 CIN 60). ” needs to be reorganized.
- Line 334: The word “quantified” should be corrected as “quantify”
- Line 367: “Statistical analyzes were performed Graph Pad Prism software” should be corrected as “Statistical analyses were performed by using GraphPad Prism software”
Author Response
The submitted manuscript entitled “Cinnamaldehyde increases the survival of mice submitted to sepsis induced by extraintestinal pathogenic Escherichia coli reducing lung lesions and systemic inflammation” describes the compound cinnamaldehyde as a promising candidate for the development of new drugs. The methods are clearly described and the paper is of interest to the readers of Antibiotics, and I would recommend accepting the manuscript after the comments below are addressed:
Line 23: “The aim of this study was” could be revised as “The study aimed to”
Line 27: The word “analyzes” should be corrected as “analysis”
Line 28: The word “was” should be corrected as “were”
Line 32: The word “cell” should be corrected as “cells”
Line 34: “…damage, decreasing…” should be corrected as “…damage by decreasing…”
Line 35: “for the development of”
Answer: We would like to thank all comments and suggestions, which greatly improved the quality of the manuscript. All suggestions, without exception, were made in the manuscript. The modifications are highlighted in yellow in the text.
Line 41: The word “problem” should be corrected as “problems”
Answer: Thank you very much for your observation. Although, in this sentence, “problem” is singular because is related with sepsis.
Line 60: “As an example”
Line 65: add the word “and” before “anti-inflammatory properties”
Answer: These adjustments have been made in the text.
Lines 66-68: What is this sentence trying to describe? Please double-check.
Answer: This sentence has been rewritten. (lines 62-64)
“Several functions have been described for the compound, such as: anticancer/antitumor activity [17], cardioprotective effect [18], anti‐inflammatory properties [19,20], and antimicrobial activity against several pathogens, such as fungi [21] and bacteria [16,22].”
Line 84: Add a comma between “added” and “and”
Line 88: “treated with cinnamaldehyde at the concentration of 60 mg/kg” or “treated with the dose”
Line 101: The word “dosis” should be corrected as “dose”
Line 125: “with the presence of edema”
Line 128: “to the non-treated infected group”
Line 147: “promotes an increase of”
Line 163: “to the non-treated infected”
Line 177: The word “cell” should be corrected as “cells”
Line 178: “to the non-treated infected”
Line 213: Remove the extra “7”
Line 222: “cinnamaldehyde for two hours…”
Line 288: Is there any unit of “density of 0.1 at 600 nm.”
Line 289: “in concentrations ranged from” should be corrected as “at concentrations ranging from”
Line 290: “was added resazurin 0.03% to” should be corrected
Line 302: “plates were incubated for 3 h”
Answer: We have modified the text as suggested. The modifications are highlighted in yellow in the manuscript.
Line 318: the sentence “finally, two therapeutic groups that were infected with E. coli F5 and after 2 h 318 received cinnamaldehyde at 30 mg/Kg (Therapeutic CIN 30) or 60 mg/kg (Therapeutic 319 CIN 60).” needs to be reorganized.
Answer: This sentence was reorganized. (Lines 313-314)
“Finally, two therapeutic groups that received cinnamaldehyde at 30 mg/Kg (Therapeutic CIN 30) or 60 mg/kg (Therapeutic CIN 60) 2 h after infection.”
Line 334: The word “quantified” should be corrected as “quantify”
Line 367: “Statistical analyzes were performed Graph Pad Prism software” should be corrected as “Statistical analyses were performed by using GraphPad Prism software”
Answer: We have modified the text as suggested.
Reviewer 2 Report
In the article, the authors evaluate the in vivo efficacy of cinnamaldehyde as an antibiotic in mice induced with pathogenic E. coli. The authors perform detailed analysis of the bactericidal effect, cytotoxicity, survival rate, histopathology, and blood cell count in the mice upon treatment with cinnamaldehyde compared to PBS control. Based on these results, they conclude that post-infection treatment with a high dose of cinnamaldehyde prevented the death of 40% of the mice studied compared to none in control. Although the result is of interest, there were only moderate differences in the tissue damage caused between treated and control group. Similarly, the count of lymphocytes, neutrophils, and monocytes were similar in the two infected groups. There was limited increase in the peritoneal cells and bone marrow cell count along with measurable decrease in the cytokines.
Overall, the article is thorough, and the discussion provides the literature basis for many of their conclusions. For example, the authors surmise that the 40% survival rate is not because of an actual decrease in the bacterial load, but because of a reduction in the inflammatory mediators. However, on line 240, the authors claim that cinnamaldehyde increased the number of lymphocytes, which according to Fig 5 is not the case. The only result where a significant effect of cinnamaldehyde can be seen is in reduction of the cytokines. As a result, although there was a reduction in mortality, it is not clear if cinnamaldehyde has any in vivo antibiotic activity. Perhaps additional study into the mode of in vitro bactericidal activity can shed light on whether it is a viable candidate as an antibiotic.
Author Response
“In the article, the authors evaluate the in vivo efficacy of cinnamaldehyde as an antibiotic in mice induced with pathogenic E. coli. The authors perform detailed analysis of the bactericidal effect, cytotoxicity, survival rate, histopathology, and blood cell count in the mice upon treatment with cinnamaldehyde compared to PBS control. Based on these results, they conclude that post-infection treatment with a high dose of cinnamaldehyde prevented the death of 40% of the mice studied compared to none in control. Although the result is of interest, there were only moderate differences in the tissue damage caused between treated and control group. Similarly, the count of lymphocytes, neutrophils, and monocytes were similar in the two infected groups. There was limited increase in the peritoneal cells and bone marrow cell count along with measurable decrease in the cytokines.”
Answer: In fact, a moderate reduction in tissue damage, a limited increase in the peritoneal and bone marrow cell counts, and a decrease in the number of cytokines, were sufficient to increase animal survival by 40%. It is important to emphasize that all these changes, even moderate, were obtained with only 8 h of treatment and one dose of the compound. E. coli F5 causes lethal sepsis within 12 hours and the cinnamaldehyde was able to quickly improve the critical and lethal condition of the animals. In the course of sepsis, where patients may die within a few hours, an increase in survival with just one dose of the compound is a very important result and of great value.
“Overall, the article is thorough, and the discussion provides the literature basis for many of their conclusions. For example, the authors surmise that the 40% survival rate is not because of an actual decrease in the bacterial load, but because of a reduction in the inflammatory mediators. However, on line 240, the authors claim that cinnamaldehyde increased the number of lymphocytes, which according to Fig 5 is not the case. The only result where a significant effect of cinnamaldehyde can be seen is in reduction of the cytokines.”
Answer: The reviewer is correct pointing out the reduction of lymphocytes in the bloodstream. In this way, the text has been adjusted for a better understanding:
“Previously, we described that Pic reduces the number of T and B lymphocytes and compromises the expression of several cell surface molecules in leukocytes, such as CD80 and CD86 [7]. Probably, this decreasing of blood cell and peritoneal cell numbers is related to a failure in the recruitment caused by Pic, since in the bone marrow the production was not compromised.” (lines 235-239)
“Here, we showed that cinnamaldehyde induced an increase of leukocytes in the bone marrow, and spleen, as well as in the peritoneum, the focus of infection. On the other hand, the treatment with cinnamaldehyde did not alter the cell profile in the blood in the infected mice. The same was observed by Mendes et al. [29] investigating the effects of cinnamaldehyde in the inflammatory response triggered by LPS injection in mice. The study showed that cinnamaldehyde treatment did not affect the number of total peripheral blood leukocytes or the number of circulating polymorphonuclear cells in LPS-injected mice.” (Lines 241-248)
“As a result, although there was a reduction in mortality, it is not clear if cinnamaldehyde has any in vivo antibiotic activity. Perhaps additional study into the mode of in vitro bactericidal activity can shed light on whether it is a viable candidate as an antibiotic.”
Answer: The reviewer is correct. In fact, here we showed that the compound presented an in vitro activity against E. coli F5. However, recently we evaluated the role of cinnamaldehyde in controlling intestinal colonization of animals infected with pathogenic enteroaggregative E. coli 042 (https://doi.org/10.3390/biom11020302). In this work, cinnamaldehyde drastically reduced intestinal infection, showing that the compound not only has action in vitro, but also in vivo, giving support to this antibiotic action against E. coli infections. In addition, Pereira et al. (2021) evaluated the action of the compound over 15 days, which is enough period to observe an antimicrobial effect. Here, the period used to evaluate the cinnamaldehyde effect was shorter, considering that, the animals were euthanized within 10 hours and the compound was administered via gavage only 2 hours of infection. We believe that the period between the absorption of the compound and the euthanasia of the animals was a limitation to observe a bactericidal effect. It is also important to mention the severity and lethality in this sepsis model.
Reviewer 3 Report
The paper should be the continuation of Pereira et al 2021 (Biomolecules 2021, 11, 302. https://doi.org/10.3390/biom11020302. ) but it presents some similarities with this one. The paper should only focus on its effects on blood, peritoneal lavage, lung, liver, spleen, kidney and cytokines.
Author Response
“The paper should be the continuation of Pereira et al 2021 (Biomolecules 2021, 11, 302. https://doi.org/10.3390/biom11020302) but it presents some similarities with this one. The paper should only focus on its effects on blood, peritoneal lavage, lung, liver, spleen, kidney and cytokines.”
Answer: Thank you very much for the reviewer´s observation. Both articles assess the role of cinnamaldehyde in infections caused by E. coli. In the paper by Pereira et al. (2021) we evaluated the effect of the compound on diarrheagenic E. coli (enteroaggregative E. coli, EAEC) and its role in intestinal colonization. In the present study, we evaluated the role of cinnamaldehyde in sepsis. In fact, the reviewer is correct mentioning that both articles present some similarity. However, although the studies share the cytotoxicity and MIC/MBC assays, Pereira used the MIC and MBC to evaluate the effect against other E. coli strains, such as diarrheagenic E. coli. In this article we used the MIC and MBC to evaluate the action of the compound against an extraintestinal E. coli, which causes sepsis. On the other hand, with regard to cytotoxicity, even performed in different periods of time, we used the same cell line and the same concentrations of the compound. Thus, we accepted the suggestion and this result was removed from the manuscript.
Reviewer 4 Report
The manuscript of Figueiredo et al. entitled “Cinnamaldehyde increases the survival of mice submitted to sepsis induced by extraintestinal pathogenic Escherichia coli reducing lung lesions and systemic inflammation” described effect of cinnamaldehyde, which is the main active constituent of cinnamon essential oil, on the experimental sepsis in mice caused by an extraintestinal pathogenic strain of E. coli. The authors demonstrated that cinnamaldehyde at a dose of 60 mg/kg was able to keep 40% of mice alive after infection that was due to immunological properties of the drug. The treatment significantly reduced the levels of cytokines in serum and peritoneum, increased the production of cell in both bone marrow and spleen, and lymphocytes at the infection site. Cinnamaldehyde was able to reduce the tissue damage, decreasing the deleterious effects for the organism and seems to be a promising candidate for development of new drugs.
The paper is well designed, well written and carefully prepared.
However, there are some comments, according to which a number of corrections should be made to the text of the article.
- There is some discrepancy between the title of the article and its main text. The emphasis on reducing lung damage in the text is not so strong to make it into the heading of the article. So, the mention of reducing lung damage in the title of the article does not seem entirely justified. The title of the article, in my opinion, needs to be changed and possibly shortened.
- It is necessary to clarify the method of preparation of the drug for both MIC experiments and animal experiments with the obligatory indication of the solvents used.
- The text at the beginning of the paragraph (lines 174 - 178) "The decreasing of blood cell and peritoneal cell numbers in the infected mice was not related to a decreasing in the cell production by bone marrow, since there was no difference among infected and control groups" is not quite clear and needs to be clarified.
- The fact that the administration of cinnamaldehyde to infected animals leads to an increase in the number of total bone marrow cells, while its administration to uninfected animals reduces the number of these cells (Fig. 7), in my opinion, needs to be explained.
Author Response
“The manuscript of Figueiredo et al. entitled ‘Cinnamaldehyde increases the survival of mice submitted to sepsis induced by extraintestinal pathogenic Escherichia coli reducing lung lesions and systemic inflammation’ described effect of cinnamaldehyde, which is the main active constituent of cinnamon essential oil, on the experimental sepsis in mice caused by an extraintestinal pathogenic strain of E. coli. The authors demonstrated that cinnamaldehyde at a dose of 60 mg/kg was able to keep 40% of mice alive after infection that was due to immunological properties of the drug. The treatment significantly reduced the levels of cytokines in serum and peritoneum, increased the production of cell in both bone marrow and spleen, and lymphocytes at the infection site. Cinnamaldehyde was able to reduce the tissue damage, decreasing the deleterious effects for the organism and seems to be a promising candidate for development of new drugs.
The paper is well designed, well written and carefully prepared.
However, there are some comments, according to which a number of corrections should be made to the text of the article.”
“There is some discrepancy between the title of the article and its main text. The emphasis on reducing lung damage in the text is not so strong to make it into the heading of the article. So, the mention of reducing lung damage in the title of the article does not seem entirely justified. The title of the article, in my opinion, needs to be changed and possibly shortened.”
Answer: As suggested, the title was modified to: “Cinnamaldehyde increases the survival of mice submitted to sepsis induced by extraintestinal pathogenic Escherichia coli”.
“It is necessary to clarify the method of preparation of the drug for both MIC experiments and animal experiments with the obligatory indication of the solvents used.”
Answer: Thank you very much for your observation. Cinnamaldehyde used in this study was obtained commercially from Sigma-Aldrich and 20% DMSO was used as solvent in the assays. As suggested, this information was added in the manuscript. (Lines 298-300)
“Cinnamaldehyde (trans-Cinnamaldehyde 99%) used in this study was obtained commercially from Sigma-Aldrich (Darmstadt, Germany). To carry out the experiments, 20% dimethyl sulfoxide (DMSO) was used as solvent.”
“The text at the beginning of the paragraph (lines 174 - 178) ‘The decreasing of blood cell and peritoneal cell numbers in the infected mice was not related to a decreasing in the cell production by bone marrow, since there was no difference among infected and control groups’ is not quite clear and needs to be clarified.”
Answer: We agree with the reviewer; this part of the text is confusing. To clarify, an alteration was made and some comments were added to the discussion, as described below:
“The decreasing of blood cell and peritoneal cell numbers in the infected mice was not related to a decreasing in the cell production by bone marrow, considering that no differences among infected and PBS groups were observed in bone marrow cells counting (Fig. 6).” (lines 166-169)
“Previously, we described that Pic reduces the number of T and B lymphocytes and compromises the expression of several cell surface molecules in leukocytes, such as CD80 and CD86 [7]. Probably, this decreasing of blood cell and peritoneal cell numbers is related to a failure in the recruitment caused by Pic, since in the bone marrow the production was not compromised.” (lines 235-239)
“Here, we showed that cinnamaldehyde induced an increase of leukocytes in the bone marrow and spleen, as well as in the peritoneum, the focus of infection. On the other hand, the treatment with cinnamaldehyde did not alter the cell profile in the blood in the infected mice. The same was observed by Mendes et al. [29] investigating the effects of cinnamaldehyde in the inflammatory response triggered by LPS injection in mice. The study showed that cinnamaldehyde treatment did not affect the number of total peripheral blood leukocytes or the number of circulating polymorphonuclear cells in LPS-injected mice.” (lines 241-248)
“The fact that the administration of cinnamaldehyde to infected animals leads to an increase in the number of total bone marrow cells, while its administration to uninfected animals reduces the number of these cells (Fig. 7), in my opinion, needs to be explained.”
Answer: We agree with the reviewer. It is possible to observe a small reduction in the number of cells in the uninfected group. However, this reduction is not statistically significant. On the other hand, after treatment of infected animals with cinnamaldehyde, there is a significant increase in the number of cells, not only in the bone marrow, but also in the spleen. These comments were added to the manuscript:
“It is possible to observe a small reduction in the number of cells in the uninfected group. However, this reduction is not statistically significant. On the other hand, the administration of cinnamaldehyde to infected animals leads to a significant increase in the number of total cells in both bone marrow and spleen (Fig. 7) when compared to the non-treated infected group.” (Lines 178-182)
Round 2
Reviewer 2 Report
The authors have duly addressed the concerns raised, making the publication of interest to the readers of Antibiotics.